# Case Study Using Recommended Reference Genes *Actin* and *18S* for Reverse-Transcription Quantitative Real-Time PCR Analysis in *Myzus persicae*

**Saqib Rahman**[1], **Zhenzhen Zhao**[1], **Muhammad Umair Sial**[2], **Yanning Zhang**[1☯], **Hongyun Jiang**[1☯]*

**1** State Key Laboratory for Biology of Plant Diseases and Insect Pests, Institute of Plant Protection, Chinese Academy of Agricultural Sciences, Beijing, PR China, **2** Institute of Plant Protection, Muhammad Nawaz Shareef University of Agriculture, Multan, Pakistan

☯ These authors contributed equally to this work.
* ptnpc@vip.163.com

**Data Availability Statement:** All relevant data are within the paper and its Supporting Information files.

## Abstract

*Myzus persicae* is a globally important pest with the ability to adjust to a wide range of environmental situations, and many molecular technologies have been developed and applied to understand the biology and/or control this pest insect directly. Reverse-transcription quantitative real-time PCR (RT-qPCR) is a primary molecular technology that is used to quantify gene expression. Choosing a stable reference gene is significantly important for precisely clarifying the expression level of the target gene. *Actin* and *18S* have been recommended as stable compounds for real-time RT-qPCR in *M. persicae* under the tested biotic and abiotic conditions. In this study, we checked the stability of *Actin* and *18S* by analyzing the relative expression levels of the cytochrome 450 monooxygenase family member genes *CYP6CY3* and *CYP6-1*, carboxylesterase gene *E4* and vacuolar protein sorting gene *VPS11* via RT-qPCR under various conditions. The expression levels of these four target genes were normalized using both *Actin* and *18S* individually and the combination of these two genes. Our results confirmed that *Actin* and *18S* can be used as reference genes to normalize the expression of target genes under insecticide treatment and starvation in *M. persicae*. However, at the developmental stages of *M. persicae*, the expression of the four tested target genes was normalized stably by *Actin* but not *18S*, with the latter presenting a problematic change with the developmental stages. Thus, the stability of reference genes in response to diverse biotic and abiotic factors should be evaluated before each RT-qPCR experiment.

## Introduction

*Myzus persicae* is commonly known as green peach aphid and represents the most well-known and dangerous agricultural pest worldwide. This pest induces injury to plants via direct feeding

**Funding:** This work was financially supported by the National Key Research and Development Program of China (2016YFD0200500). The funders had no role in study design, data collection and analysis, decision to publish, or preparation of the manuscript.

**Competing interests:** The authors have declared that no competing interests exist.

and spreading various plant viruses; thus, it leads to severe yield losses in economically important crop commodities [1, 2]. For the control of this insect, many molecular technologies have been developed and applied to understand the biology of *M. persicae* and/or control this pest insect directly [3–5].

Reverse-transcription quantitative real-time PCR (RT-qPCR) is a primary molecular technology used to quantify gene expression [6]. Typically, this technique requires the systematic normalization of the expression level of a target gene with one or more reference genes as internal controls to account for the different gene input quantities due to the different RNA samples [7]. Housekeeping genes are defined as genes expressed ubiquitously in all cell types and tissues regardless of the physiological status, development phase and treatment, and they should be easily detected and expressed at a constant level; consequently, they are used extensively as reference genes for RT-qPCR analysis [7, 8]. However, a single housekeeping gene that can satisfy all given experimental conditions has not been identified. Housekeeping genes are tissue-specific and condition-dependent, and these characteristics greatly affect the reliability and accuracy of RT-qPCR analysis [8, 9]. The use of unsuitable reference genes in RT-qPCR has been shown to lead to false results and then to erroneous interpretations [10–14]. Choosing the most stable reference gene in a specific tissue and treatment is significantly important to precisely clarify the expression level of the target gene [14, 15]. To date, more than one reference gene is required for accurate normalization in RT-qPCR analysis [8, 9].

In insect species, many experiments have been conducted in gene expression studies to select and evaluate reference genes for RT-qPCR analysis [16–19]. With the number of analysis tools, experimental factors and reference genes as parameters, the current trends in the selection of reference genes in insects were reviewed systematically based on a total of 90 representative papers, which were published mainly between 2013 and 2017 [12]. This review indicates that it is necessary to conduct reference gene screening experiments or validate the reasonability of reference genes used in each RT-qPCR. In addition, reference genes were selected and evaluated from a total of 78 insect species, including *M. persicae*, according to the 90 cited papers [12, 19]. In that publication, the stability of nine housekeeping genes, namely, *18S ribosomal RNA* (*18S*), *ribosomal protein gene* (*RPL27*, *RPL7* and *RPL32*), *glyceraldehyde-3-phosphate dehydrogenase* (*GAPDH*), *acetylcholinesterase* (*ACE*), *β-tubulins*, *Actin* and *elongation factor 1 alpha* (*EF*1A), was evaluated comprehensively with one method and three software under various biotic and abiotic conditions [19]. Among those nine reference genes, *Actin* and *18S* were confirmed as the most stable for RT-qPCR in *M. persicae* under all tested biotic and abiotic conditions (development stage, tissue, host post, wing dimorphism, insecticide, photoperiod and temperature) [19]. However, the suitability of these two reference genes may fluctuate according to the specific tissues and conditions.

In this case study, to validate the two reference genes *Actin* and *18S* recommended by Kang et al. in *M. persicae* [19], we analyzed the relative expression levels of four target genes, including the two cytochrome 450 monooxygenase family member genes *CYP6CY3* and *CYP6-1*, carboxylesterase gene *E4* and vacuolar protein sorting gene *VPS11*, with RT-qPCR analysis under both biotic and abiotic conditions. CYP6CY3 and E4 are belong to metabolic enzymes, which have been studied widely and deeply and associated to insecticide resistance in *M. persicae* [1]. The over-expression of these two genes induced by insecticide treatment even as sublethal doses have been observed in many insects including *M. persicae* [20–22]. CYP6-1 is a member of cytochrome 450 monooxygenase family. VPS11 is involved in vesicle transport to vacuoles which plays an important role in segregation of intracellular molecules into distinct organelles [23]. However, both CYP6-1 and VPS11 are unfamiliar in *M. persicae*. The changes in the expression pattern of these two genes induced by the changes of both biotic and abiotic conditions would be unpredictable, which will give the relative comparison. Additionally, the

expression levels of these four target genes were normalized using both *Actin* and *18S* individually and the two-gene combination with the $2^{-\Delta\Delta Ct}$ method.

## Material and methods

### Insects

A field population of *M. persicae* collected from Nanping, Fujiang Province in 2018 was used in this study. Aphids were grown on the leaves of Chinese cabbage (*Brassica napus L var chinensis*) in an incubator without the use of any pesticides at 22 ± 2˚C, 65 ± 5% relative humidity and a 16 Light: 8 Dark photoperiod [1, 3].

### Insecticidal stress

Two insecticides, flonicamid 97.1% and pymetrozine 98%, were used to treat *M. persicae* in this research, which are normally useful in the control of *M. persicae*. According to the bioassay results for these two insecticides (data not shown), three concentrations of flonicamid (17.2, 34.4 and 68.8 mg/L) and pymetrozine (10.0, 20.0 and 40.0 mg/L), which induced sublethal effects, were used to treat adults of *M. persicae* with the leaf dip method 01]. In the control group, Chinese cabbage leaves were treated with distilled water containing 1% EtOH. A total of 30 adults were collected to expose each dose of the two tested insecticides under standard conditions, 22 ± 2˚C, 65 ± 5% relative humidity and a 16 Light: 8 Dark photoperiod. Three replicates were conducted for each treatment, and mortality was recorded after 24 h.

### Starvation treatments

To create starvation stress, a group of 30 newly emerged adults of *M. persicae* as one treatment were put into glass petri dishes without food for a certain amount of time, including 6, 12, 24 or 48 hours, and placed in a climatic chamber under controlled conditions [1]. The aphids fed Chinese cabbage leaves corresponding to a certain time were used as the control group. All treatments were replicated three times, and the relevant mortality was recorded.

### Aphid developing phases

Five different growing stages of *M. persicae* (green peach aphids), including 1st, 2nd, 3rd, and 4th instar nymphs and newly emerged adults, were collected as samples [24]. For each nymph sample, a total of 60 aphids were collected, and for each adult sample, a total of random 30 aphids were collected. All samples were collected three times.

### RNA isolation and cDNA synthesis

A total of 20–30 adult or 50–60 nymph *M. persicae* from all treatments were assembled per biological replication to isolate RNA with RNeasy® Mini Kits (Qiagen, ON, Canada) according to the manufacturer's protocol. A NanoDrop ND_1000 (NanoDrop Technologies, Wilmington, DE, USA) was used to conduct both qualitative and quantitative analyses. RNA integrity was confirmed by evaluating the samples by gel electrophoresis (1% gel). RNA samples with an absorbance ratio range (OD) of 260/280 between 1.90 and 2.20 were used in further studies [24, 25]. Primary stand cDNA was created from 1 μg of the total RNA in a 20 μl volume by using the Omniscript® Reverse Transcript Kit (Qiagen, ON, Canada) and then stored at -20˚C for future analysis.

**Table 1. Primers used in real-time qRT-PCR.**

| Genes | Primer sequences (5'-3') | Length (bp) | PCR efficiency (%) | $R^2$ | Origin of primers |
|---|---|---|---|---|---|
| *CYP6CY3* | F: CGGGGTGACGATCATCTATT | 128 | 94.2 | 0.983 | Accession number: HM009309 |
| | R: GGGTGGTCTTTTGACAAAGC | | | | |
| *CYP6-1* | F: TCAACGAATGTGGCGACGAA | 120 | 102 | 0.982 | Accession number: AJ310561 |
| | R: CGCAGGTGGCAATTACGTCT | | | | |
| *E4* | F: AAACTTTCCTTTTACACCGTT | 160 | 100 | 0.991 | Accession number: X74554 |
| | R: TCTAAGCCAAGAAATGTTGAAA | | | | |
| *VPS11* | F: GGGTCAAGGAATAACTGGAATGGC | 180 | 105 | 0.991 | Accession number: XM_022313476 |
| | R: TCAGGTCCATTCACTAACACCGA | | | | |
| *Actin* | F: GGTGTCTCACACACAGTGCC | 142 | 100 | 0.996 | Bass et al., 2011 [22] |
| | R: CGGCGGTGGTGGTGAAGCTG | | | | |
| *18S* | F: TCAACACGGGAAACCTCACCA | 80 | 96.6 | 0.992 | Coleman et al., 2015 [23] |
| | R: CACCACCCACCGAATCAAGAA | | | | |

$R^2$, regression coefficient.

## Gene expression with RT-qPCR

Primers used for the RT-qPCR analysis of the target genes *CYP6CY3*, *CYP6-1*, *E4* and *VPS11* were designed with online software (http://www.idtdna.com/scitools/applications/primerquest/) based on sequences in the National Center for Biotechnology Information (NCBI) database (Table 1). The primers used for the reference genes *Actin* and *18S* are also shown in Table 1 and were cited from published previous studies [26, 27]. The amplification efficiency for each pair of specific primers was detected by the slope of the standard curve generated using a series of 10-fold dilutions of cDNAs [25].

RT-qPCR was evaluated using TransStart® Top Green qPCR SuperMix (Transgen, Beijing, China) with QuantStudio 3 Real-Time PCR Systems (Applied Biosystems, USA). According to the manufacturer's instructions, the whole 20 $\mu$L reaction mixture was completed in a 50 $\mu$L tube containing 10 $\mu$L of 2 × TransStart® Top Green qPCR SuperMix, 0.4 $\mu$L of each primer, 8.2 $\mu$L of distilled ddH$_2$O, and 1 $\mu$L of cDNA template. The RT-qPCR conditions were 30 seconds at 94˚C, followed by 40 cycles at 94˚C for 30 seconds and then annealing at 72˚C for 30 seconds. The RT-qPCR study involved three self-determining biological replicates for each treatment. The relative expression of the four tested target genes was calculated with the *Actin* and *18S* individually as reference genes according to Pfaffl [6] and with the two used together as reference genes according to the methods developed by Vandesompele et al. (2002) [8].

## Single control normalization error *E*

According to a report by Vandesompele et al. (2002) [8], the possible errors of using only one reference gene to normalize the expression levels of the target gene, termed single control normalization errors (*Es*), were calculated.

## Statistical analysis

Data were statistically evaluated using ANOVA and monitored by the LSD test in SPSS software version 22.0. The results were considered significant when the *P* value was < 0.05. For the expression of relative quantities of genes, Tukey's comparison was used to test the mean differences between treatments across generations and separated by LSD tests.

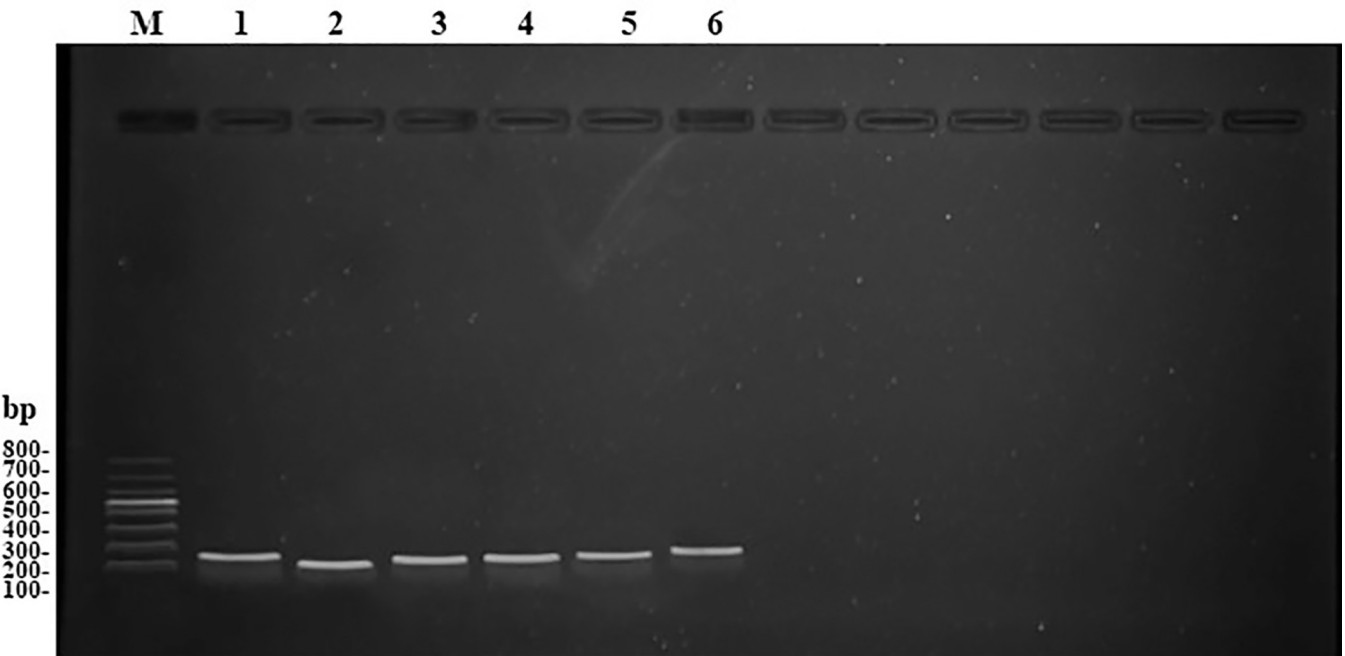

**Fig 1. Gel electrophoresis analysis of PCR amplification for each target and reference gene.** M, 100 bp DNA Ladder (Takara Bio INC, Beijing, China); 1–6, *Actin*, *18S*, *CYP6CY3*, *CYP6-1*, *E4* and *VPS11*.

## Results

### Amplification efficiency of PCR

As shown in Fig 1, a unique PCR product detected by 2% agarose gel electrophoresis analysis ensured that a single amplicon was produced for each target and reference gene. The PCR efficiency and regression coefficient ($R^2$) were evaluated by the establishment of standard curves. As shown in Table 1, the efficiency values of all primers ranged from 94.2 to 105% with regression coefficient values from 0.98 to 1.00, which reached the threshold of 90.0 to 110% for the PCR efficiency and 0.98 to 1.00 for the regression coefficient ($R^2$) [24, 28]. These results suggested that all the primer pairs were appropriate for the RT-qPCR analysis.

### Expression of target genes normalized to *Actin* and *18S* under the insecticide treatment

After exposure for 24 hours with flonicamid and pymetrozine by the leaf dip method, the mortality of *M. persicae* was less than 10.0% for the two tested insecticides, even at the highest doses of 68.8 mg/L for flonicamid and 40.0 mg/L for pymetrozine (Table 2). In the RT-qPCR

**Table 2. Mortality of *M. persicae* after exposure to different doses of flonicamid and pymetrozine for 24 hours.**

| Flonicamid | | Pymetrozine | |
|---|---|---|---|
| Concentrations (mg/L) | Mortality (%) | Concentrations (mg/L) | Mortality (%) |
| 0 | 0 | 0 | 0 |
| 17.2 | 5.55 ± 3.84 | 10.0 | 4.44 ± 3.84 |
| 34.4 | 8.89 ± 1.92 | 20.0 | 6.67 ± 3.33 |
| 68.8 | 10.0 ± 3.33 | 40.0 | 8.89 ± 3.84 |

assay, the fold expression of four target genes was normalized using *Actin* and *18S* as the internal control. When *M. persicae* was treated with flonicamid, remarkable changes in the four target genes (*CYP6CY3*, *CYP6-1*, *E4* and *VPS11*) induced by insecticide treatment were observed with both *Actin* and *18S* as reference genes. A comparison of the expression data normalized to *Actin* and *18S* showed that a similar tendency occurred for *CYP6CY3* and *VPS11* with different concentrations of flonicamid (Fig 2A and 2B) while significant differences occurred between *CYP6-1* and *E4* at flonicamid doses of 17.2 mg/L and 68.8 mg/L, respectively. For pymetrozine, the relative expression levels showed a similar expression pattern for *CYP6CY3*, *CYP6-1* and *VPS11* with either *Actin* or *18S* used alone for normalization, and *CYP6CY3* and *VPS11* were obviously increased at the highest dose of 40.0 mg/L (Fig 2D and 2E). However, the relative expression level of *E4* was increased significantly when normalized to *Actin* but not to *18S* at the highest dose of 40.0 mg/L pymetrozine. In addition, the relative expression levels of target genes were also normalized using the combination of *Actin* and *18S* (Fig 2C and 2F). The results showed that similar expression profiles were observed for all four tested genes compared with those normalized with *18S* alone (Fig 2B, 2C, 2E and 2F).

## Expression of the target genes normalized to *Actin* and *18S* in *M. persicae* after starvation

As shown in Fig 3, the tendency of expression changes of the four target genes normalized to *Actin* was generally consistent with those normalized to *18S* and was initially inhibited and then promoted with changes in starvation time (Fig 3A and 3B). After starvation for 6 h, the relative expression levels of four target genes were all inhibited significantly compared to those without starvation treatment (starvation for 0 hours) normalized both with *Actin* and *18S* as reference gene respectively, except for *E4* and *VSP11* normalized with *Actin*. The relative expression levels of four target genes were all increased significantly in *M. persicae* after starvation for 48 h normalized both with *Actin* and *18S* as reference gene respectively. A comparison

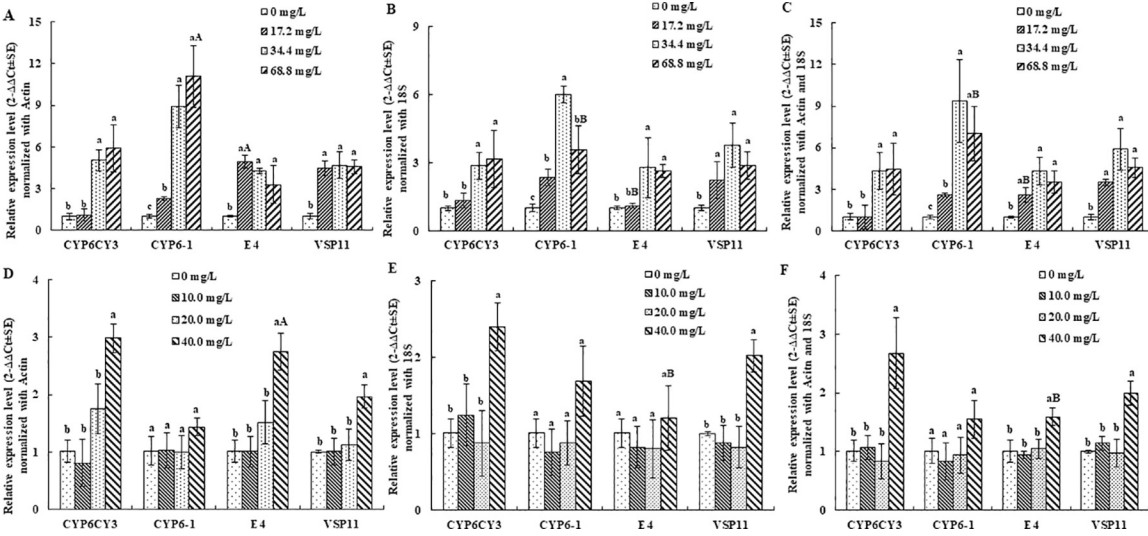

**Fig 2. Fold expression of four target genes normalized to *Actin* and *18S* under insecticide treatment.** A, B, C, Relative expression levels normalized to *Actin* and *18S* alone and in combination under the flonicamid treatment; and D, E, F, relative expression levels normalized to *Actin* and *18S* alone and in combination under the pymetrozine treatment. Values are shown as the mean ± SEM. Bars with different letters are significantly different from each other (*P* < 0.05). Lowercase letters indicate the comparison among different concentrations for one target gene normalized with the same housekeeping gene. Uppercase letters represent the comparison among different normalized genes for one target gene under the same treatment of one insecticide.

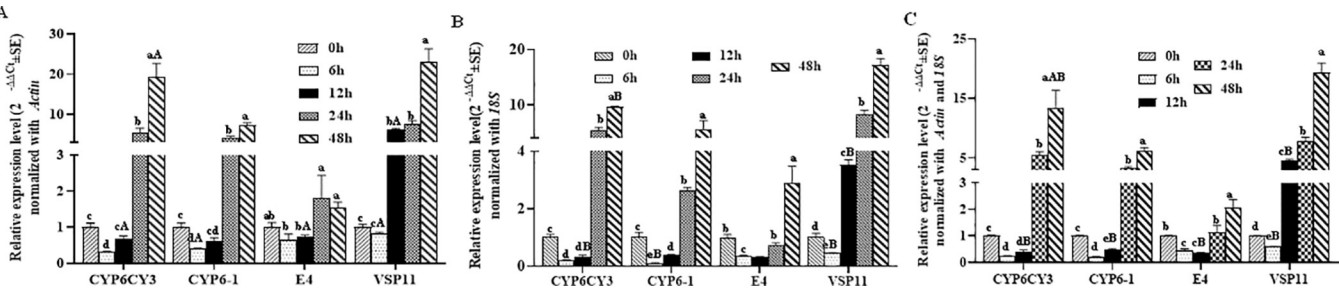

**Fig 3.** Relative expression levels of target genes in *M. persicae* after starvation normalized to *Actin* (A) and *18S* (B) alone and in combination (C). Values are shown as the mean ± SEM. Bars with different letters are significantly different from each other (*P* < 0.05). Lowercase letters indicate the comparison among different starvation time for one target gene normalized with the same housekeeping gene. Uppercase letters represent the comparison among different normalized genes for one target gene under the same starvation time.

between the relative expression levels normalized with *Actin* and *18S* showed that significant differences occurred for all tested target genes, with *CYP6CY3* at 12 h and 48 h, *CYP6-1* at 6 h, *E4* at 12 h, and *VPS11* at 6 h and 12 h. As shown in Fig 3C, the expression fold normalized to the combination of *Actin* and *18S* showed no differences compared with those normalized to *18S* individually (Fig 3B and 3C).

## Expression of target genes normalized to *Actin* and *18S* at different developmental stages of *M. persicae*

With the 1st instar nymphs of *M. persicae* as the control group, the expression levels of the four target genes (*CYP6CY3*, *CYP6-1*, *E4* and *VPS11*) at different developmental stages of *M. persicae* are shown in Fig 4, which illustrates that the tendency of expression changes of the four target genes with the developmental stage were obviously different based on the normalization to *Actin* or *18S* (Fig 4A and 4B). When normalized to *Actin* alone, the relative expression ratios of the four tested target genes changed in a certain regular pattern with the developmental stages of *M. persicae*. However, the fold change in expression of the four tested genes normalized to *18S* alone fluctuated irregularly with the developmental stage in *M. persicae*. The mean threshold cycle (*Ct*) and standard error values of *Actin* and *18S* in all samples tested for the developmental stages were 16.5 ± 1.5 and 14.4 ± 4.8, respectively. However, the mean *Ct* and standard error values of *Actin* and *18S* in all samples tested for the insecticide treatment and starvation conditions were 19.6 ± 0.3 and 18.3 ± 2.0 and 12.9 ± 0.9 and 11.7 ± 1.3, respectively. Additionally, we calculated the stability of *Actin* and *18S* with the

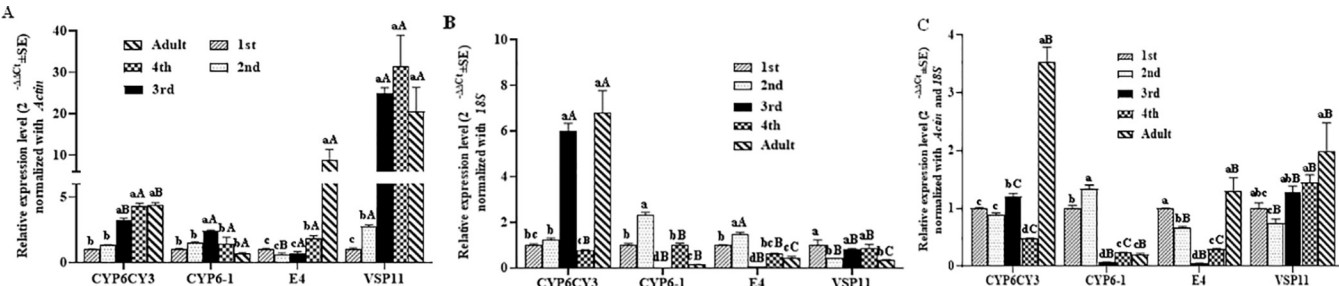

**Fig 4.** Expression of target genes at different developmental stages of *M. persicae* normalized to *Actin* (A) and *18S* (B) alone and in combination (C). Values are shown as the mean ± SEM. Bars with different letters are significantly different from each other (*P* < 0.05). Lowercase letters indicate the comparison among different developmental stages for one target gene normalized with the same housekeeping gene. Uppercase letters represent the comparison among different normalized genes for one target gene under the same developmental stage.

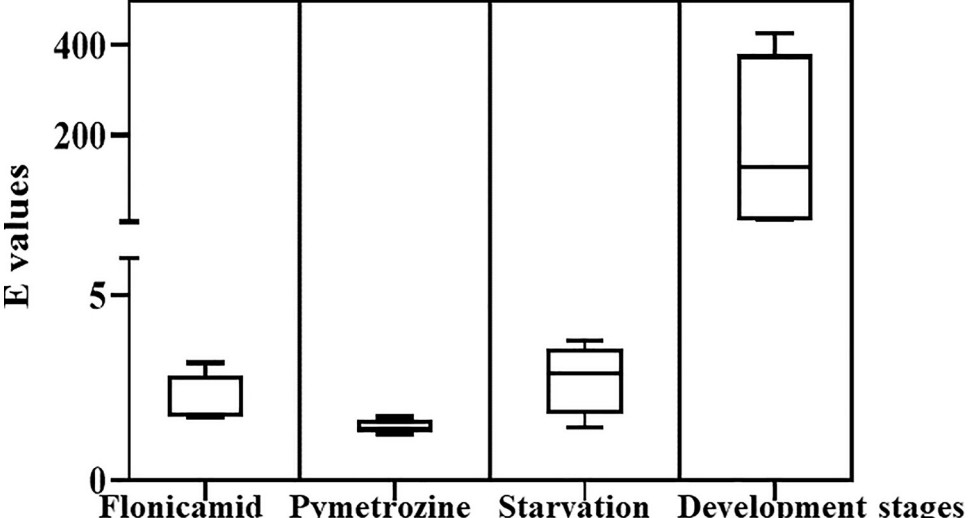

**Fig 5. Single control normalization error *E* for four target genes under different biotic and abiotic conditions in *M. persicae*.** The boxes indicate the 25th and 75th percentiles. Lines across the boxes represent the medians, and upper and lower dots represent the maximum and minimum values, respectively.

software program NormFinder at the different developmental stages of *M. persicae* [29]. This result indicated that *Actin* (0.87) was more stable than *18S* (2.06) in this subset. The relative expression levels of all four tested target genes normalized to *Actin* and *18S* together are shown in Fig 4C, which indicated that correction could not be made due to the instability of *18S*.

## Single control normalization error *E*

To evaluate the possible errors induced by using only *Actin* or *18S* as a reference gene to normalize the relative expression levels of the four tested target genes, we calculated the single control normalization error *E* according to the method reported by Vandesompele et al. (2002) [8]. Under the insecticide treatment, the *E* values indicating the ratio of the fold expression between normalization to *Actin* and *18S* ranged from 1.68 (for *VSP11*) to 3.17 (for *CYP6-1*) under flonicamid treatment and from 1.25 (for *E4*) to 1.71 (for *CYP6-1*) under pyrethrozine treatment (Fig 5). Under starvation conditions, the minimum *E* value was 1.42 for *VSP11* and the maximum *E* value was 3.78 for *E4*. However, the *E* value during the developmental stages reached a maximum of 426 for *CYP6-1* and a minimum of 6.58 for *VSP11* (Fig 5).

## Discussion

RT-qPCR was developed and is frequently used to determine messenger RNA expression levels elaborated in many biological procedures [30, 31]. Different types of reference genes are used as internal controls to normalize the expression of target genes between different samples in many experiments [14, 24]. However, the expression of a specific reference gene changes under different environments and treatments, which can affect the real expression of a particular target gene. Therefore, a stable endogenous reference gene is essential for relative quantification in gene expression analyses. Currently, many reference genes have been recommended in different insect species, including sweet potato whitefly *Bemisia tabaci* (Gennadius) [32–34], brown planthopper *Nilaparvata lugens* (Stål) [35, 36], diamondback moth *Plutella xylostella* (Linnaeus) [15, 17], fruit fly *Drosophila melanogaster* (Meigen) [37], pea aphid *Acyrthosiphon pisum* (Harris) [38] and peach aphid *M. persicae* (Sülzer) [19]. According to the report of

Kang et al. (2017) [19], *18S* and *Actin* were recommended as the most stable reference genes in *M. persicae* for abiotic (photoperiod, temperature and insecticide susceptibility) and biotic (host plant, developmental stage, tissues and wing dimorphism) studies. In this paper, *18S* and *Actin* were used to confirm their stability as reference genes for normalizing the selected four target genes *CYP6CY3*, *CYP6-1*, *E4* and *VPS11* under different treatments, including insecticide, starvation and development.

*Actin* encodes a major structural protein and is expressed at various levels in many cell types. It is regarded as an ideal internal control and widely used as a reference gene for RT-qPCR in many organisms. Recently, the stability of *Actin* as a housekeeping gene has been investigated frequently in insects [12, 39]. *Actin* is the most stable reference gene, including *ribosomal protein L* (*RPL*), *tubulin*, *GAPDH*, *ribosomal protein S* (*RPS*), *18S*, *EF1A*, *TATA box binding protein* (*TATA*), *heat shock protein* (*HSP*) and *succinate dehydrogenase complex subunit A* (*SDHA*), across different developmental stages of many insects, including honeybee *Apis mellifera* (Linnaeus) [16], diamondback moth *P. xylostella* (Linnaeus) [15, 17], fruit fly *D. melanogaster* (Meigen) [37], striped stem borer *Chilo suppressalis* (Walker) [40], etc. [12, 39]. Previous studies have indicated that *Actin* is less stable in several members of the Coccinellidae family, including the seven-spotted lady beetle *Coccinella septempunctata* (Linnae) [41], ladybird beetle *Coleomegilla maculata* (DeGeer) [42] and convergent lady beetle *Hippodamia convergens* (Guérin-Méneville) [43]. *18S* ribosomal RNA is a part of the ribosomal RNA (rRNA), which accounts for > 80% of the total RNA pool [12, 13], whereas mRNA accounts for only 3–5%. Several recent studies concluded that the use of ribosomal RNA as an endogenous control in a variety of cellular systems is consistently the best choice compared with other methods [39, 44]. However, it is generally acknowledged that the use of rRNA forms as reference genes to normalize the expression of target genes of mRNA species is problematic due to the potential to mask subtle changes in target genes [12, 45].

In this study, the values of single control normalization error *E* for *Actin* and *18S* ranged from 1.25 to 3.78 under the two tested conditions. According to Vandesompele et al., the average 75th and 90th percentile *E* values were 3.0 (range 2.1–3.9) and 6.4 (range 3.0–10.9), respectively [8]. Additionally, the expression pattern of target genes with changes in condition was similar between the normalization of *Actin* and *18S*. These data confirmed again that *Actin* and *18S* are stable reference genes for the normalization of the expression levels of multiple genes under the starvation and insecticide treatment conditions. However, when comparing the detailed expression levels normalized with *Actin* and *18S* individually, the relative expression ratios with *Actin* as a reference gene were higher than those normalized with *18S* in some cases. The relative expression levels of the combination of *Actin* and *18S* were fully consistent with *18S* alone as a reference gene under both the insecticide and starvation treatments. According to Kang et al., the stability of nine reference genes in *M. persicae* under an insecticide treatment was ranked from the most stable to the least stable as follows: *L27* (*ribosomal protein L27*) > *18S* > *RPL7* >*EF1A*> *RPL32* > *GAPDH* > *Actin* > *β-tubulin* >*ACE* (*acetylcholinesterase*) [19]. The stability of the nine selected candidate reference genes, including *Actin* and *18S*, was not evaluated directly under starvation conditions [19]. According to our data, *Actin* and *18S* were also stable in *M. persicae* under these application scenarios, which provides formative guidance for the application of *Actin* and *18S* in *M. persicae*. Furthermore, the expression levels of the four tested target genes in *M. persicae* could be more accurately normalized with the combination of *Actin* and *18S*. Therefore, the importance of carefully selecting more than one stable and suitable reference gene for RT-qPCR normalization was suggested, although target gene expression could be normalized with either *Actin* or *18S* alone.

The developmental stage is an important experimental factor that has been widely investigated for its effect on the stability of reference genes in insects [12, 39]. In this study, the

tendency of expression changes of four target genes with the development stage obviously differed when normalized to *Actin* or *18S* individually and in combination. The mean *Ct* and standard error value value of *Actin* in all samples tested for developmental stages was 16.5 ± 1.5, and the stability calculated with the software program NormFinder was 0.87 [8]. In the same subset, the mean *Ct* and standard error and stability values for *18S* were 14.4 ± 4.9 and 2.06, respectively. The *E* value during the developmental stages reached a maximum of 426 for *CYP6-1* and a minimum of 6.58 for *VSP11*. These results indicated that *Actin* is stable in normalizing the expression of the four tested target genes in *M. persicae* at different developmental stages. However, *18S*, which has been regarded as a more stable reference gene than *Actin* in *M. persicae* [19], showed a problematic change with the development stages. Unstable reference genes either alone or in combination cannot be used to normalize the expression of target genes in the analysis of RT-qPCR result. It is recommended to evaluate the stability of reference genes in response to diverse biotic and abiotic factors before each RT-qPCR experiment.

## Acknowledgments

The field population of *M. persicae* was kindly provided by Dr. Jinfeng Hu, Institute of Plant Protection, Fujian Academy of Agricultural Sciences, Fuzhou, Fujian, PR China.

## Author Contributions

**Investigation:** Saqib Rahman, Muhammad Umair Sial.

**Writing – original draft:** Zhenzhen Zhao.

**Writing – review & editing:** Yanning Zhang, Hongyun Jiang.

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
