## [Decision Letter · Decision Letter 0]

18 Mar 2021

PONE-D-21-00308

Case Study Using Recommended Reference Genes Actin and 18S for Reverse-Transcription Quantitative Real-Time PCR Analysis in Myzus persicae

PLOS ONE

Dear Dr. Jiang,

Thank you for submitting your manuscript to PLOS ONE. After careful consideration, we feel that it has merit but does not fully meet PLOS ONE’s publication criteria as it currently stands. Therefore, we invite you to submit a revised version of the manuscript that addresses the points raised during the review process.

I agree with the comments proposed by the two reviewers. Please improve the manuscript according to their suggestions.

We look forward to receiving your revised manuscript.

Kind regards,

Yonggen Lou

Academic Editor

PLOS ONE

Journal Requirements:

2)  Thank you for stating the following in the Acknowledgments Section of your manuscript:

[This work was financially supported by the National Key Research and Development Program of China (2016YFD0200500).]

 [The funders had no role in study design, data collection and analysis, decision to

publish, or preparation of the manuscript.]

3)  PLOS ONE now requires that authors provide the original uncropped and unadjusted images underlying all blot or gel results reported in a submission’s figures or Supporting Information files. This policy and the journal’s other requirements for blot/gel reporting and figure preparation are described in detail at https://journals.plos.org/plosone/s/figures#loc-blot-and-gel-reporting-requirements and https://journals.plos.org/plosone/s/figures#loc-preparing-figures-from-image-files. When you submit your revised manuscript, please ensure that your figures adhere fully to these guidelines and provide the original underlying images for all blot or gel data reported in your submission. See the following link for instructions on providing the original image data: https://journals.plos.org/plosone/s/figures#loc-original-images-for-blots-and-gels.

Reviewers' comments:

Reviewer's Responses to Questions

**Comments to the Author**

1. Is the manuscript technically sound, and do the data support the conclusions?

Reviewer #1: Yes

Reviewer #2: Partly

2. Has the statistical analysis been performed appropriately and rigorously? 

Reviewer #1: Yes

Reviewer #2: Yes

3. Have the authors made all data underlying the findings in their manuscript fully available?

Reviewer #1: Yes

Reviewer #2: No

4. Is the manuscript presented in an intelligible fashion and written in standard English?

Reviewer #1: Yes

Reviewer #2: Yes

5. Review Comments to the Author

Reviewer #1: The manuscript written by Rahman et al. reported that actin but not 18s can be used to analyze gene expression at development stages of Myzus persicae. This is a basic work but is very important for studying molecular biology of M. persicae. Overall, the experiments were well designed and the manuscript was well written.

I only have a few minor concerns:

1) In page 4, line 3, five methods should be revised as one method and three software.

2) In figure 2, 3 and 4, please put “Actin”, “18S”, “Actin and 18S” into each panel. This will make the figures clear.

3) In page 9, line 18, you should give the full name when first used.

4) In page 17, line 24, “under the same treatment of one insecticide”?

Reviewer #2: This manuscript by Rahman et al. validated the stability of two reference genes Actin and 18S by analyzed the relative expression levels of four target genes. I have some questions for this research.

First, it is better to standardize the nomenclature associated with quantitative PCR to avoid confusion; for example, the abbreviation RT-PCR should be used for reverse transcription PCR, qPCR should be used for quantitative real-time PCR, and RT-qPCR should be used for reverse transcription-qPCR. However, in this study, the abbreviations relative to PCR is very confusing; there are four types of abbreviation, "real-time qRT-PCR", "real-time RT-qPCR", "RT-qPCR" and "qRT-PCR".

Second, since the purpose of this study is to validate the stability of two reference genes, it is necessary to show all the Ct values if possible.

Third, consider to rearrange Figure 2-4. It is hard for reader to compare the tendency of the relative expression levels of one target gene (normalized to Actin, 18S, or two-gene combination). I think it is better to put the three results of one same gene (normalized to Actin, 18S, or two-gene combination) into an independent line chart. Then combine the four graphs (four target genes that with the same treatment) together.

Fourth, it is not nice for reviewer to make comment when the line number of each page is independent.

In addition, the followings should be considered in revising the manuscript.

Page 2, line 13: The initial of actin should be capitalized. Please check the full manuscript.

Page 3, line 9: quantify gene expression

Page 4,

Line1: the appropriate abbreviation of ribosomal protein L27 should be RPL27. And RPL32 should be italic.

Line 10-12: why you chose this four target genes has to be mentioned in introduction.

Line 21: need format modification. Please check the full manuscript.

Page 5,

Line 11: Adult aphids used in the study are female only, male only, or random. And how many days have the adults emerged.

Line 24, 25: The cDNAs have been synthesized by reverse transcription PCR, hence, only qPCR here. It has to be clarified if the "qRT-PCR" here stands for "quantitative real-time PCR".

Page 6,

Line 9: Pfaffl [6]

Line 26-28: It is better to count the numbers to same decimal places.

Page 7,

Line 3: The mortalities were too low and have no differences among three concentrations. It might be exposure to insecticides only for 24 h or the concentrations of these two insecticides might be not appropriate. Besides, the mortality data of control was not listed in Table 2.

Line 10, 26: I noticed that the differences only occurred at one or two tested time point, and not all target genes show differences when normalized with Actin, 18S, or two-gene combination. It seems that the difference occur relative to the expression profile of different target gene or the deviation among samples. These results were not persuasive to evaluate the stability of the reference gene.

Page 8,

Line 8: Define Ct. Besides, for example, 16.51±1.51 indicated mean ± standard error or standard deviation.

Page 9,

Line 18: Define RPS, TATA, HSP and SDHA.

Page 10,

Line 12: The reference genes were ranked from the most stable to the least stable, but the "<" here was very confusing when lack of enough background. I think it is not appropriate to copy exactly the description in the reference.

6. PLOS authors have the option to publish the peer review history of their article (what does this mean?). If published, this will include your full peer review and any attached files.

Reviewer #1: No

Reviewer #2: No

---

## [Author Response · Author response to Decision Letter 0]

2 May 2021

Response to Reviewer 1

Reviewer #1: The manuscript written by Rahman et al. reported that actin but not 18s can be used to analyze gene expression at development stages of Myzus persicae. This is a basic work but is very important for studying molecular biology of M. persicae. Overall, the experiments were well designed and the manuscript was well written. I only have a few minor concerns:

Response:

 We are very grateful to your comments on the manuscript. At the same time, we have thoroughly revised the manuscript in order to express the idea more clearly. We believe the manuscript has been greatly improved. Once again, thank you for the kind advice and acknowledgments.

1. In page 4, line 3, five methods should be revised as one method and three software.

Response:

DONE.

“…five methods…” was changed to “one method and three software”.

2. In figure 2, 3 and 4, please put “Actin”, “18S”, “Actin and 18S” into each panel. This will make the figures clear.

Response:

DONE.

“Actin”, “18S” and “Actin and 18S” were added into each panel.

3. In page 9, line 18, you should give the full name when first used.

Response:

DONE.

The full name was added for each gene when it was used firstly, as follow. “…..ribosomal protein L (RPL), Tublin, GADPH, ribosomal protein S (RPS), 18S, EF1A, TATA box binding protein (TATA), heat shock protein (HSP) and succinate dehydrogenase complex subunit A (SDHA)”.

4. In page 17, line 24, “under the same treatment of one insecticide”?

Response:

DONE.

 Page 14, lines 15-18, “among different concentrations…under the same treatment of one insecticide…” was changed to “…..among different starvation time…..under the same starvation time”. Page 14, lines 23-26, “among different concentrations…under the same treatment of one insecticide…” was changed to “among different developmental stages ….under the same developmental stage”.

Response to Reviewer 2

Reviewer #2: This manuscript by Rahman et al. validated the stability of two reference genes Actin and 18S by analyzed the relative expression levels of four target genes. I have some questions for this research.

Response:

We are very grateful to your comments on the manuscript. At the same time, we have thoroughly revised the manuscript in order to express the idea more clearly. We believe the manuscript has been greatly improved. Once again, thank you for the kind advice and acknowledgments.

1. First, it is better to standardize the nomenclature associated with quantitative PCR to avoid confusion; for example, the abbreviation RT-PCR should be used for reverse transcription PCR, qPCR should be used for quantitative real-time PCR, and RT-qPCR should be used for reverse transcription-qPCR. However, in this study, the abbreviations relative to PCR is very confusing; there are four types of abbreviation, "real-time qRT-PCR", "real-time RT-qPCR", "RT-qPCR" and "qRT-PCR".

Response:

DONE.

We checked this manuscript carefully and standardized the nomenclature associated with quantitative PCR to only one type, RT-qPCR.

2. Second, since the purpose of this study is to validate the stability of two reference genes, it is necessary to show all the Ct values if possible.

Response:

 DONE.

 The mean Ct and standard error values of of Actin and 18S for all samples in this paper were shown in the section of “Expression of target genes normalized to Actin and 18S at different developmental stages of M. persicae”.

3. Third, consider to rearrange Figure 2-4. It is hard for reader to compare the tendency of the relative expression levels of one target gene (normalized to Actin, 18S, or two-gene combination). I think it is better to put the three results of one same gene (normalized to Actin, 18S, or two-gene combination) into an independent line chart. Then combine the four graphs (four target genes that with the same treatment) together.

Response:

DONE.

Thanks for your good suggestion. We reviewed Figures 2-4 according to the suggestions of Reviewer 1. We added “Actin”, “18S”, “Actin and 18S” into each panel.

4. Fourth, it is not nice for reviewer to make comment when the line number of each page is independent.

Response:

DONE. 

We have numbered each line of this paper continuously. 

In addition, the followings should be considered in revising the manuscript.

5. Page 2, 

Line 13: The initial of actin should be capitalized. Please check the full manuscript.

Response:

DONE.

 We have checked carefully the full manuscript and corrected.

6. Page 3, line 9: quantify gene expression.

Response:

DONE.

“….quantity RNA expression…” was changed to “….quantity gene expression…”.

7. Page 4,

Line1: the appropriate abbreviation of ribosomal protein L27 should be RPL27. And RPL32 should be italic.

Response:

DONE.

 “L27” was changed to “RPL27” and RPL32 was in italic type.

Line 10-12: why you chose this four target genes has to be mentioned in introduction.

Response:

DONE.

We have given the reasons as follow: CYP6CY3 and E4 are belong to metabolic enzymes, which have been studied widely and deeply and associated to insecticide resistance in M. persicae [1]. The over-expression of these two genes induced by insecticide treatment even as sublethal doses have been observed in many insects including M. persicae [20-22]. CYP6-1 is a member of cytochrome 450 monooxygenase family. VPS11 is involved in vesicle transport to vacuoles which plays an important role in segregation of intracellular molecules into distinct organelles [23]. However, both CYP6-1 and VPS11 are unfamiliar in M. persicae. The changes in the expression pattern of these two genes induced by the changes of both biotic and abiotic conditions would be unpredictable, which would give a relative comparison.

Line 21: need format modification. Please check the full manuscript.

Response:

DONE.

“….a 16 L:8D photoperiod…” was changed to “….a 16 Light: 8 Dark photoperiod…”. And We check the full manuscript carefully. 

8. Page 5,

Line 11: Adult aphids used in the study are female only, male only, or random. And how many days have the adults emerged.

Response:

DONE.

This section was change to “Five different growing stages of M. persicae (green peach aphids), including 1st, 2nd, 3rd, and 4th instar nymphs and newly emerged adults, were collected as samples [20]. For each nymph sample, a total of 60 aphids were collected, and for each adult sample, total random 30 aphids were collected.”

Line 24, 25: The cDNAs have been synthesized by reverse transcription PCR, hence, only qPCR here. It has to be clarified if the "qRT-PCR" here stands for "quantitative real-time PCR".

Response:

DONE.

We checked this manuscript carefully and standardized the nomenclature associated with reverse transcription PCR and quantitative real-time PCR.

9. Page 6,

Line 9: Pfaffl [6]

Response:

DONE.

“Pfaffl6” was changed to “Pfaffl [6]”.

Line 26-28: It is better to count the numbers to same decimal places.

Response:

DONE. 

The data was unified to the same decimal places.

10. Page 7,

Line 3: The mortalities were too low and have no differences among three concentrations. It might be exposure to insecticides only for 24 h or the concentrations of these two insecticides might be not appropriate. Besides, the mortality data of control was not listed in Table 2.

Response:

DONE. 

Firstly, the mortality data of control group was listed in Table 2. Secondly, the population used in this study is a field population collected from Nanping, Fujiang Province in 2018 (Sial et al., 2020, [3]), which have developed a certain level of resistance to pesticides. Therefore, the concentrations of tested two insecticides used this study just induced sublethal effects. And the highly mortalities were not the aim of this study. The metabolic resistance mechanism related to the over-production of carboxylesterases E4 and Cytochrome P450 monooxygenases CYP6CY3 seem to confer low to moderate levels of resistance to pesticides in Myzus persicae, which have been documented in a lot of paper. In a word, the changes in the expression of metabolic enzymes induced by sublethal-doses of pesticides were detected in this paper.

Line 10, 26: I noticed that the differences only occurred at one or two tested time point, and not all target genes show differences when normalized with Actin, 18S, or two-gene combination. It seems that the difference occur relative to the expression profile of different target gene or the deviation among samples. These results were not persuasive to evaluate the stability of the reference gene.

Response:

DONE. 

This section was changed to “As shown in Fig 3, the tendency of expression changes of the four target genes normalized to Actin was generally consistent with those normalized to 18S and was initially inhibited and then promoted with changes in starvation time (Figs 3A and B). After starvation for 6h, the relative expression levels of four target genes were all inhibited significantly compared to those without starvation treatment (starvation for 0 hours) normalized both with Actin and 18S as reference gene respectively, except for E4 and VSP11 normalized with Actin. The relative expression levels of four target genes were all increased significantly in M. persicae after starvation for 48h normalized both with Actin and 18S as reference gene respectively. A comparison between the relative expression levels normalized with Actin and 18S showed that significant differences occurred for all tested target genes, with CYP6CY3 at 12 h and 48 h, CYP6-1 at 6 h, E4 at 12 h, and VPS11 at 6 h and 12 h. As shown in Fig 3C, the expression fold normalized to the combination of Actin and 18S showed no differences compared with those normalized to 18S individually (Figs 3B and C).”

11. Page 8,

Line 8: Define Ct. Besides, for example, 16.51±1.51 indicated mean ± standard error or standard deviation.

Response:

DONE. 

The define “threshold cycle” was added for Ct. And “….the Ct values….” were changed to “mean threshold cycle (Ct) and standard error values” in lines 8 and 10. 

12. Page 9,

Line 18: Define RPS, TATA, HSP and SDHA.

Response:

DONE. 

The full name for RPS, TATA, HSP and SDHA were added. 

13. Page 10,

Line 12: The reference genes were ranked from the most stable to the least stable, but the "<" here was very confusing when lack of enough background. I think it is not appropriate to copy exactly the description in the reference.

Response:

DONE. 

It was changed to “>”.

---

## [Decision Letter · Decision Letter 1]

25 Jun 2021

PONE-D-21-00308R1

Case Study Using Recommended Reference Genes Actin and 18S for Reverse-Transcription Quantitative Real-Time PCR Analysis in Myzus persicae

PLOS ONE

Dear Dr. Jiang,

Thank you for submitting your manuscript to PLOS ONE. After careful consideration, we feel that it has merit but does not fully meet PLOS ONE’s publication criteria as it currently stands. Therefore, we invite you to submit a revised version of the manuscript that addresses the points raised during the review process.

Please improve the manuscript according to the suggestion of the reviewer #2.

We look forward to receiving your revised manuscript.

Kind regards,

Yonggen Lou

Academic Editor

PLOS ONE

Journal Requirements:

Reviewers' comments:

Reviewer's Responses to Questions

**Comments to the Author**

1. If the authors have adequately addressed your comments raised in a previous round of review and you feel that this manuscript is now acceptable for publication, you may indicate that here to bypass the “Comments to the Author” section, enter your conflict of interest statement in the “Confidential to Editor” section, and submit your "Accept" recommendation.

Reviewer #1: All comments have been addressed

Reviewer #2: All comments have been addressed

2. Is the manuscript technically sound, and do the data support the conclusions?

Reviewer #1: Yes

Reviewer #2: Yes

3. Has the statistical analysis been performed appropriately and rigorously? 

Reviewer #1: Yes

Reviewer #2: Yes

4. Have the authors made all data underlying the findings in their manuscript fully available?

Reviewer #1: Yes

Reviewer #2: No

5. Is the manuscript presented in an intelligible fashion and written in standard English?

Reviewer #1: Yes

Reviewer #2: Yes

6. Review Comments to the Author

Reviewer #1: (No Response)

Reviewer #2: The followings should be still considered in revising the manuscript.

1. Line 169 and 173, add a space between number and "h".

2. Line 206, italicize the "E".

3. Line 207, there is no Fig 5 in this manuscript.

4. Line 277, delete the space between "-" and "q".

7. PLOS authors have the option to publish the peer review history of their article (what does this mean?). If published, this will include your full peer review and any attached files.

Reviewer #1: No

Reviewer #2: No

---

## [Author Response · Author response to Decision Letter 1]

28 Jul 2021

Response to Reviewer 2

Reviewer #2: The followings should be still considered in revising the manuscript.

Response:

 We are very grateful to your comments on the manuscript. At the same time, we have thoroughly revised the manuscript in order to express the idea more clearly. We believe the manuscript has been greatly improved. Once again, thank you for the kind advice and acknowledgments.

1. Line 169 and 173, add a space between number and "h".

Response:

DONE. A space was added between number and “h”. 

2. Line 206, italicize the "E".

Response:

DONE. The “E”was changed to “E”.

3. Line 207, there is no Fig 5 in this manuscript.

Response:

DONE. The Fig 5 was submitted.

4. Line 277, delete the space between "-" and "q".

 DONE. The space was deleted between "-" and "q".

---

## [Editor Report · Decision Letter 2]

22 Sep 2021

Case Study Using Recommended Reference Genes Actin and 18S for Reverse-Transcription Quantitative Real-Time PCR Analysis in Myzus persicae

PONE-D-21-00308R2

Dear Dr. Jiang,

We’re pleased to inform you that your manuscript has been judged scientifically suitable for publication and will be formally accepted for publication once it meets all outstanding technical requirements.

Kind regards,

Yonggen Lou

Academic Editor

PLOS ONE
---

## [Editor Report · Acceptance letter]

8 Oct 2021

PONE-D-21-00308R2 

Case Study Using Recommended Reference Genes *Actin* and 18S for Reverse-Transcription Quantitative Real-Time PCR Analysis in *Myzus persicae*

Dear Dr. Jiang:

I'm pleased to inform you that your manuscript has been deemed suitable for publication in PLOS ONE. Congratulations! Your manuscript is now with our production department. 

Kind regards, 

on behalf of

Dr. Yonggen Lou 

Academic Editor

PLOS ONE